# Predicting Non-Small-Cell Lung Cancer Survival after Curative Surgery via Deep Learning of Diffusion MRI

**DOI:** 10.3390/diagnostics13152555

**Published:** 2023-08-01

**Authors:** Jung Won Moon, Ehwa Yang, Jae-Hun Kim, O Jung Kwon, Minsu Park, Chin A Yi

**Affiliations:** 1Department of Radiology, Kangnam Sacred Heart Hospital, Hallym University School of Medicine, Seoul 07441, Republic of Korea; jwmooooon@gmail.com; 2Department of Radiology, Samsung Medical Center, Sungkyunkwan University School of Medicine, Seoul 06351, Republic of Korea; ehwayang@gmail.com; 3Division of Respiratory and Critical Care Medicine, Department of Internal Medicine, Samsung Medical Center, Sungkyunkwan University School of Medicine, Seoul 06351, Republic of Korea; ojung.kwon@samsung.com; 4Department of Information and Statistics, Chungnam National University, Daejeon 34134, Republic of Korea; minsu.park51@gmail.com

**Keywords:** NSCLC, MR, DWI, prognosis prediction, AI, deep learning

## Abstract

Background: the objective of this study is to evaluate the predictive power of the survival model using deep learning of diffusion-weighted images (DWI) in patients with non-small-cell lung cancer (NSCLC). Methods: DWI at b-values of 0, 100, and 700 sec/mm^2^ (DWI_0_, DWI_100_, DWI_700_) were preoperatively obtained for 100 NSCLC patients who underwent curative surgery (57 men, 43 women; mean age, 62 years). The ADC_0-100_ (perfusion-sensitive ADC), ADC_100-700_ (perfusion-insensitive ADC), ADC_0-100-700_, and demographic features were collected as input data and 5-year survival was collected as output data. Our survival model adopted transfer learning from a pre-trained VGG-16 network, whereby the softmax layer was replaced with the binary classification layer for the prediction of 5-year survival. Three channels of input data were selected in combination out of DWIs and ADC images and their accuracies and AUCs were compared for the best performance during 10-fold cross validation. Results: 66 patients survived, and 34 patients died. The predictive performance was the best in the following combination: DWI_0_-ADC_0-100_-ADC_0-100-700_ (accuracy: 92%; AUC: 0.904). This was followed by DWI_0_-DWI_700_-ADC_0-100-700_, DWI_0_-DWI_100_-DWI_700_, and DWI_0_-DWI_0_-DWI_0_ (accuracy: 91%, 81%, 76%; AUC: 0.889, 0.763, 0.711, respectively). Survival prediction models trained with ADC performed significantly better than the one trained with DWI only (*p*-values < 0.05). The survival prediction was improved when demographic features were added to the model with only DWIs, but the benefit of clinical information was not prominent when added to the best performing model using both DWI and ADC. Conclusions: Deep learning may play a role in the survival prediction of lung cancer. The performance of learning can be enhanced by inputting precedented, proven functional parameters of the ADC instead of the original data of DWIs only.

## 1. Introduction

Lung cancer is the most common cause of cancer death, accounting for 26.6% of all cancer deaths [1]. The survival of lung cancer patients is expected differently according to the stage of lung cancer when it is diagnosed. Non-small-cell lung cancer (NSCLC) with localized disease without regional or distant metastasis shows 59.0% 5-year relative survival, whereas NSCLC with distant metastasis shows only a 5.8% 5-year relative survival rate [2]. The possibility of survival of lung cancer at the time of diagnosis can only be estimated using an alleged percentage of survival at each given stage of lung cancer. The stage of lung cancer is determined based on the extent of the primary cancer and the extent of metastasis through lymphatic drainage and hematogenous metastasis or pleural seeding, which can be determined using anatomic information obtained via preoperative CT, PET/CT, or MRI, and confirmed via percutaneous biopsy, surgical biopsy, or curative surgery [3].

MRI is a state-of-the-art imaging modality which can picturize human anatomy with good contrast and resolution. Although the ability of pulmonary nodule detection is lower than CT, MRI can detect malignant nodules without radiation exposure. In the TNM staging of lung cancer, MRI shows superiority in the evaluation of mediastinal or chest wall invasion, or Pancoast tumor, and comparable N staging assessment capability with the use of PET/CT. The use of diffusion-weighted images (DWIs) achieves good results of detection and characterization of malignant nodule or lymph node metastasis [4]. On DWIs, MRI can quantify functional information such as cellular density and molecular movability in a tissue. The anatomic information from MRI can be interpreted using human visual perception, but functional parameters of MRI, such as diffusion and perfusion parameters, are presented with numbers for each pixel, and its clinical significance cannot be inferred visually. The apparent diffusion coefficient (ADC) value reveals the diffusivity of water molecules and can be quantified from functional indices [5,6]. The measured ADC value can be used in the discrimination of the benign or malignant nature of tumors using the criterion of the optimal cutoff of 1.470 × 10^−3^ mm^2^/s in lung cancer and the subtype differentiation of renal cell carcinoma [7,8]. It is also helpful in prognosis evaluation in terms of chemotherapy response prediction in malignancies such as breast cancer, rectal cancer, osteosarcoma, and hepatic metastasis from colorectal cancer [9,10,11,12]. Concerning prognosis prediction, lower values of the ADC suggest more aggressive histologic types and grades, and worse prognosis [13,14,15], whereas there is also a report that states suspicion in the usefulness of ADC values in determining the histological grade of malignancy, despite the excellence of MRI staging in endometrial cancer [16].

Artificial intelligence (AI) is a mechanism of computing based on learning and thinking from data itself, imitating human behavior. Machine learning, a system of learning through experience, is a subset of AI, and deep learning is also a subset of machine learning, which represents systems based on neural networks. The convolutional neural network (CNN) is a class of methods which was successfully applied in image analysis such as detection, classification, segmentation, and prediction tasks using medical image data such as pathology, X-ray, CT, and MRI. Medical images are also quite a promising field of research, using CNN in the detection and classification of pathology and the prediction of clinically relevant outcomes [17,18,19,20,21,22,23,24,25,26,27,28]. To our knowledge, there have been no reports of deep learning using CNN in the analysis of DWI and the prognosis of lung cancer. Deep learning from diffusion MRI may clarify the clinical significance of functional information which is encoded in DWI and ADC, which cannot be analyzed upon viewing diffusion-weighted images. The problem that we primarily identified in previous survival prediction models was that previous models [29,30,31,32] have only utilized clinical data (e.g., age, sex, smoking history) in their survival prediction models, without using diffusion MRI. We have not only incorporated conventional MRI images into our model (DWI_0_), but we have also incorporated the functional parameters of diffusion MRI (ADC), since diffusion MRI can be analyzed more accurately when its functional parameters of the ADC are used alongside the images themselves. Functional parameters naturally lend themselves well to deep learning, because each of the functional parameters assigned to each pixel—which are extremely numerous in aggregate—can readily be optimized using deep learning.

The purpose of this study is to evaluate the predictive power of prognostic model learning from DWI only or learning from both DWI and ADC or DWI, ADC, and clinical information in patients with NSCLC.

## 2. Materials and Methods

### 2.1. Patients

The institutional review board of our institution approved this study as a part of a clinical trial for the staging of lung cancer, which was registered as a randomized clinical trial with ClinicalTrials.gov number NCT01065415. Written informed consent was obtained from all patients in the single tertiary referral hospital. From January 2010 through to November 2011, patients with stage I, II, or IIIA NSCLC (other than N2 disease) based on clinical staging underwent conventional work up including physical examination, laboratory tests, bronchoscopy, chest CT, or PET/CT upon admission (*n* = 151). In cases of an inappropriate condition for surgery, such as poor pulmonary function, poor performance status (ECOG 3 or 4), concurrent medical diseases, history of malignancy treatment, contraindication for MR image acquisition, or refusal of involvement, patients were excluded (*n* = 51). After MR image acquisition, thoracotomy with or without mediastinoscopy was performed, and 100 patients were included (57 men, 43 women; mean age, 62 years).

We evaluated age, sex, smoking history, tumor size, pathologic type, surgical stage of NSCLC (AJCC 7th), and survival information based on an electrical chart review. The causes of death statistics were updated annually by the National Statistical Office, and the electrical charts of cancer patients had their updated survival information. From the date of MR acquisition, 5-year survival was determined by the date of death or last follow-up date of survivors on the chart.

### 2.2. MR Acquisition

All thoracic MR examinations were performed via a 1.5-T machine (Magnetom Avanto; Siemens, Erlangen, Germany), using surface array coils. MR images were obtained with diffusion-weighted images using a single-shot, spin echo, echo planar imaging (EPI) sequence with spectrally adiabatic inversion recovery (SPAIR) fat suppression (FS) and b-values of 0, 100, and 700 s/mm^2^ (repetition time (TR)/echo time (TE) = 11,700 ms/73 ms; number of repetition averages, 4; matrix size = 192 × 162; in-plane resolution = 2.08 × 2.08 mm; FOV = 400 × 325 mm^2^; slice thickness = 5 mm; number of slices = 60).

### 2.3. Image Processing

DWI is used for the calculation of the ADC. To generate the perfusion-insensitive ADC by eliminating the pseudo-diffusion effect, the ADC was calculated based on a b-value of 100 and 700 (ADC_100-700_). The perfusion-sensitive ADC value was calculated using a b-value of 0 and 100 (ADC_0-100_). The overall conventional ADC value was calculated using a b-value of 0, 100, and 700 (ADC_0-100-700_). Specifically, the ADC value was calculated using a mono-exponential model [33]:S_(b)_/S_0_ = exp (−b × ADC),
where S_(b)_ is the signal intensity at a particular b-value, S_0_ is the signal intensity with b = 0 s/mm^2^, and b is the b-factor. The ADC value was estimated via linear fitting using Matlab (Mathworks, Natick, MA, USA). For each voxel, three ADC (ADC_0-100-700_, ADC_0-100_, and ADC_100-700_) values were estimated with a low b-value (slope between 0 and 100 s/mm^2^, ADC_0-100_; microperfusion-facilitated ADC), a high b-value (slope between 100 and 700 s/mm^2^, ADC_100-700_; perfusion-insensitive ADC), and overall b-values (slope between 0, 100, and 700 s/mm^2^, ADC_100-700_; conventional ADC). The tumor ROI was manually defined on the axial ADC_0-100-700_ map. The voxels ranging from 2.5% to 7.5% of the ADC_0-100-700_ values within the tumor ROI were extracted and averaged to compute the ADC_0-100-700_ value. The corresponding voxels were used to compute the ADC_0-100_ and ADC_100-700_ values.

The DWI and ADC images were normalized as input data for the value of signal intensity and the size of the pixel. The signal intensity of images was normalized into a range from 0 to 1 and all images were interpolated as 2 mm sized pixel images. The cancer was manually segmented on the ADC map by a radiologist (CAY with 20 years of experience). From the manually segmented lung cancer volume, the slice with the largest area of the lung cancer was selected as a mask. Then, the lung cancer region of the DWI and ADC images (DWI_0_, DWI_100_, DWI_700_, ADC_0-100_, ADC_100-700_, and ADC_0-100-700_) were segmented using the selected mask. In our dataset, the maximum size of the lung cancer was found to be 34 × 30 pixels. The segmented images were padded into the size of 56 × 56 pixels, and then resized to 224 × 224 pixels to be fed as an input to our deep learning model.

### 2.4. Deep Learning Model for Survival Prediction

In this paper, we propose a survival prediction model for lung cancer using deep learning with the transfer learning of VGG-16 as the backbone structure. VGG-16 is a convolutional neural network model proposed by K. Simonyan and A. Zisserman [34]. VGG-16 consists of sixteen layers: 13 convolutional layers, 2 fully connected layers, and 1 softmax layer for the output. The input of the network is three channels of images in 224 × 224 resolution. When three channels of images entered the model as input data, the feature maps of the network were generated through a convolution operation process with a combination of three channels of images. The output was the classification of 1000 objects through the softmax layer in the ImageNet dataset.

In this study, we modified the softmax layer of the VGG-16 model into a binary classification of survival and death. The architecture of the modified VGG-16 is described in Figure 1. When clinical information was added to our model, a fully connected layer was added in the latter part of the deep learning structure to evaluate the augmented performance for the prediction of survival in NSCLC patients.

We evaluated the predictive power of deep learning in three different combinations for input data: 1. DWI only, 2. DWI and ADC, and 3. DWI, ADC, and clinical information. As input data for the network, three channels of image data were selected from DWI and ADC. Combinations included DWI as anatomical data, ADC_0-100_ as perfusion-sensitive ADC, ADC_100-700_ as perfusion-insensitive ADC, and ADC_0-100-700_ as conventional ADC. The survival network could capture features related to the survival of the lung cancer patient from the input dataset through training. The output of the network was the survival probability of the lung cancer patients.

### 2.5. Implementation

The models were implemented using Tensorflow (version 1.14). The pretrained VGG16 model in ImageNet was used to obtain the initial parameters of our network. Our model was trained at the initial learning rate of 0.001 for one classification layer, two fully connected layers, and three convolutional layers until 70 epochs, and at the learning rate of 0.00001 for the fine tuning of the whole layers until 100 epochs. The total epoch was set to 170. Cross entropy was used for loss function, and the stochastic gradient descent was used as an optimizer. Data augmentation, such as flipping the x and y axis and rotation (−30~30), was performed during training. For the inputs of the model, the three channels of images were used as various combinations of the DWI (DWI_0_, DWI_100_, DWI_700_) and the ADC map (ADC_0-100_, ADC_100-700_, ADC_0-100-700_). Ten-fold cross validation was used to evaluate the survival prediction model. A total of 100 subjects were divided into 10 subsets containing 10 subjects for each subset. One subset (10 subjects) was used as the test set and nine subsets (90 subjects) were used as the training set. The accuracy of the model was reported as the average of the prediction accuracy from the 10 experiments.

### 2.6. Statistical Analysis

Several commonly reported performance metrics such as the area under the receiver operating characteristic curve (AUC), sensitivity, specificity, kappa, accuracy, and balanced accuracy were used to evaluate whether survival at 5 years could be classified using deep learning models which were trained with different sets of input data as the predictor. Here, in the confusion matrix, Cohen’s kappa is a measure of the proportion of a “true” agreement beyond that expended by chance, and the balanced accuracy was defined as the average of sensitivity and specificity to deal with the class imbalance problems [35].

To provide measurements of the uncertainty of the model’s prediction accuracy, we calculated the 95% confidence intervals (CIs) for the estimation of measurements by providing bootstrap samples with 1000 replications. When the 95% CI for a given comparison did not include zero, we concluded that there was a difference between the two models. The association between MR images and the 5-year survival of lung cancer patients was tested via logistic regression analysis, adjusted for clinical information such as age, sex, smoking history, tumor size, pathologic type, and surgical stage. The optimal cutoff was calculated using Youden’s index.

All statistical analyses were carried out using R packages (version 3.6.1; R Development Core Team, www.r-project.org, accessed on 13 May 2022) and SAS (version 9.4; SAS Institute, Cary, NC, USA). All statistical tests were two-sided with a significance level of 0.05.

## 3. Results

### 3.1. Demographics

Clinical, pathologic, and prognostic characteristics are summarized in Table 1. Pathologic diagnosis of 100 patients included adenocarcinoma (*n* = 63), squamous cell carcinoma (*n* = 32), adenosquamous carcinoma (*n* = 2), large cell neuroendocrine carcinoma (*n* = 1), pleomorphic adenocarcinoma (*n* = 1), and other NSCLC (*n* = 1). Among 100 NSCLC patients, 66 patients survived, and 34 patients died at 5-year follow up after curative surgery. Sixty-three patients had no progression and the remaining thirty-seven patients showed local recurrence (*n* = 2) or metastasis (*n* = 36).

### 3.2. Performance of the Survival Prediction Model Using DWI and ADC

The best predictive performance (92% accuracy) was achieved in the model learning from the combination of DWI_0_-ADC_0-100_-ADC_0-100-700_ input data (Table 2). The model trained with at least one ADC map showed high accuracies (87~92%). On the other hand, the models trained with only DWIs showed low accuracies (76% in DWI_0_-DWI_0_-DWI_0_ and 81% in DWI_0_-DWI_100_-DWI_700_ input data), although this model structure integrated features from each DWI’s input data (Figure 2).

When the accuracies were compared, the model trained using DWI_0_-ADC_0-100_-ADC_0-100-700_ input data showed significantly better performances than the model trained with only DWIs, but there was no significant difference between the models using at least one ADC input datum (Table 2).

Looking at the individual cases that the models accurately predicted, the model using both ADC and DWI accurately predicted 9 additional cases which were not predicted accurately by the model using only ADC, and 12 additional cases which were not predicted accurately by the model using only DWI. On the other hand, the model using both ADC and DWI did not make correct predictions in four cases which were correctly predicted using ADC only, and one case which was correctly predicted using DWI only. The three cases could not be predicted correctly by any of these three models (Figure 3).

### 3.3. Performance of the Survival Prediction Model Using DWI, ADC, and Clinical Information

When clinical information (age, sex, smoking history, tumor size, pathologic type, and surgical stage) was added to the AI-generated survival predictions using diffusion MRI, the survival prediction improved when demographic features were added to the model with only DWIs, but the benefit of clinical information was not prominent when added to the best performing model using both DWI and ADC (Table 3). The best performance (94%) was achieved with a model using DWI_0_-ADC_0-100_-ADC_0-100-700_ and all of the clinical information as input data, which was slightly better than the accuracy with a model using DWI_0_-ADC_0-100_-ADC_0-100-700_ only (92%). However, when clinical information was added to the model using DWI only (76~81% accuracies), the survival prediction was improved with more than a 7% increase in accuracies (83~89% accuracies).

## 4. Discussion

DWI and ADC of MR images reveal the diffusion capacity of water molecules and are widely used for oncologic imaging in terms of characterization, diagnosis, and prognosis prediction. Either a visual assessment of diffusion restriction by comparing the signal intensity of high- and low-b-value DWI or measuring the value less than 1.5 × 10^−3^ mm^2^/s on the ADC map may suggest poor prognosis of a patient. For example, based on these two assessments, radiologists could suggest a diagnosis of malignancy based on MR images, although the ADC range of lung cancer can vary [12,36]. Intense restriction on DWI and smaller ADC values can suggest poor prognosis in terms of higher pathologic grade, lymph node metastasis, and response to chemoradiation therapy, but there are no obvious criteria nor cutoff values for the differentiation of survival and death in each NSCLC patient [37,38]. Such identifications of diffusion restriction can help to predict the probability of better or poorer prognosis, but individual (personalized) prediction of 5-year death or survival for a specific patient cannot be achieved via visual assessment or value measurement only.

The prognostic prediction of NSCLC patients using deep learning models has been applied with several biomarkers such as radiologic, histopathologic, genetic, or molecular evidence [39,40,41,42]. In the medical field of pulmonary image analysis and prognosis prediction, several deep learning applications have been suggested in terms of chest radiograph, CT, or PET/CT. Lu et al. demonstrated that deep learning chest radiograph risk scoring could stratify the mortality risk of individuals of the Prostate, Lung, Colorectal, and Ovarian Cancer Screening Trial and National Lung Screening Trial [43]. Hosny et al. demonstrated a mortality assessment on the CT images of NSCLC patients via deep learning [44]. Also, Baek et al. visualized the U-Net algorithm of PET/CT in NSCLC patients in a prediction of survival. From our knowledge, this is the first study demonstrating the 5-year overall survival prognostication of NSCLC patients after curative surgery based on a deep learning model from DWI and ADC data of the tumor. The accuracy of this prediction was 92%, the highest in our model learning input data of DWI and ADC.

Signal intensity loss on the diffusion-sensitive sequence can be quantified by calculating ADC [45]. Based on non-linear transformation of the voxel values of each DWI, the deep learning model could generate ADC-like feature maps. However, in our study, we found that the deep learning model produced low-accuracy results using DWI images only. This could be due to the lack of training samples for weight optimization. The deep learning model could learn more efficiently when clinically significant parameters such as ADC_0-100_ and ADC_0-100-700_ were precalculated and then provided as input data, rather than it directly learning from the diffusion images. These clinically significant parametric maps enhance the predictive power of the deep learning model in cases of limited training samples. Alternatively, deep learning models trained solely with DWI_0_, DWI_100_, and DWI_700_ images seemed to make predictions based on the “black box” nature of deep learning models, whether or not the model had extracted clinically relevant ADC data from the DWIs. In the case that the models had not extracted ADC from DWI, both the accuracy and the reliability of the model declined significantly. Our solution to this problem was to directly provide ADC (alleged known functional parameters which reflect the cellular density) to the deep learning model, so that we could be assured that the deep learning model incorporated ADC in its decision making.

The survival prediction with a regression model incorporating clinical information and AI-generated predictions using diffusion MRI improved the accuracies of models using diffusion MRI. The benefit of the clinical information is prominent in the relatively low-performing deep learning model using DWIs only, but the gain was not prominent in the best-performing deep learning model using both DWIs and ADCs, which already showed high accuracies of 92%. It would be difficult to further increase this high accuracy with the limited amount of data in our current study.

Our study is limited due to the small number of datasets. To deal with this limitation, we applied three techniques for evaluating the survival prediction model. Firstly, ten-cross validation was performed. The cross validation technique could minimize the problem of overfitting that may occur with a small number of datasets [46,47]. In this study, the train and test datasets were divided into 9:1 and the validation was conducted crosswise 10 times to maximize the amount of data that could be learned out of 100 datasets. Secondly, transfer learning was adopted to handle possible problems such as over-fitting or a lack of datasets. In this study, the model was trained by reusing the parameters of the pre-trained VGG16, and the number of weights for optimization was reduced. Lastly, data augmentation was performed to train the network to avoid the overfitting problem. The data augmentation technique is a well-known approach in the generalization of a deep learning model. In this study, flip and rotation functions were used in data augmentation, and the same is detailed in the Methods section.

For the interpretation of the deep learning model, previous studies have shown promising results [48,49,50]. The class activation map (CAM), for example, provides the location information of contributing pixels within the images, allowing the CNN to predict the class of an image [50]. Using the CAM, we could understand which parts of the image had more of an effect on the final output of the deep learning model. In this study, however, we could not apply the CAM into our modified VGG 16 model, due to the limited deep learning architecture and transfer learning strategies.

## 5. Conclusions

In conclusion, deep learning may play a role in the survival prediction of lung cancer. The accuracy of results produced by the deep learning model can be enhanced by inputting precedented, proven, functional parameters of the ADC, including the raw data of DWI in survival prediction. The novelty of this paper lies not only in creating a new deep learning model, but also in our use of diffusion MRI data to predict survival in non-small-cell lung cancer patients—a clinical application that has not been attempted before in lung cancer survival prediction research.

## Figures and Tables

**Figure 1 diagnostics-13-02555-f001:**
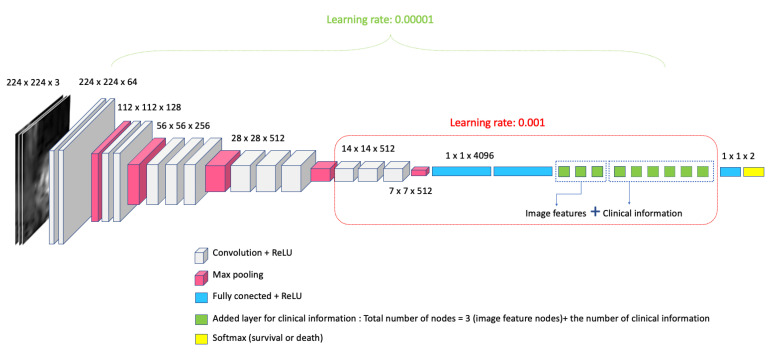
Survival prediction model architecture.

**Figure 2 diagnostics-13-02555-f002:**
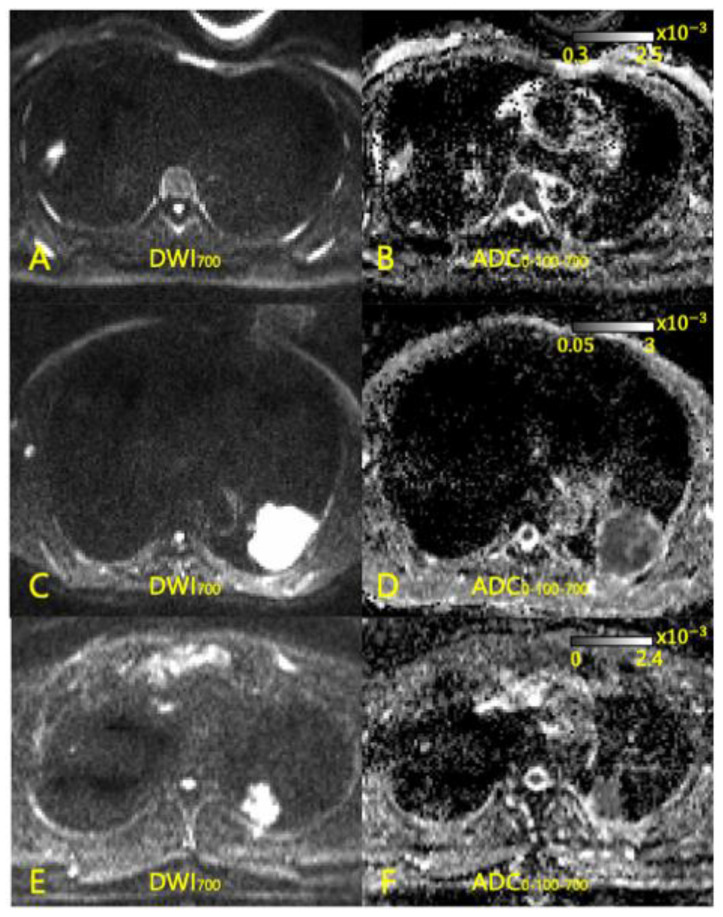
DWI and ADC images of three patients. (**A**,**B**) 50-year-old woman with T1 N0 M0 adenocarcinoma of lung. ADC value was 1.504 × 10^−3^ mm^2^/s. Our survival prediction models predicted the survival of this patient in all of the combinations of DWI and ADC. She remained alive 5 years after curative surgery. (**C**,**D**) 71-year-old man with T2 N0 M0 squamous cell carcinoma of lung. ADC value was 1.174 × 10^−3^ mm^2^/s, which is suggestive of poor prognosis. Our survival prediction models predicted death of this patient in all of the combinations of DWI and ADC. At 25 months after curative surgery, he died. (**E**,**F**) 62-year-old man with T1 N0 M0 adenocarcinoma of lung. ADC value was 1.12 × 10^−3^ mm^2^/s, which could suggest poor prognosis, but he remained alive 5 years after curative surgery. Deep learning model with DWI-only combination (DWI_0_-DWI_0_-DWI_0_, DWI_0_-DWI_100_-DWI_700_) failed to predict the survival, but the model with ADC input predicted his survival correctly.

**Figure 3 diagnostics-13-02555-f003:**
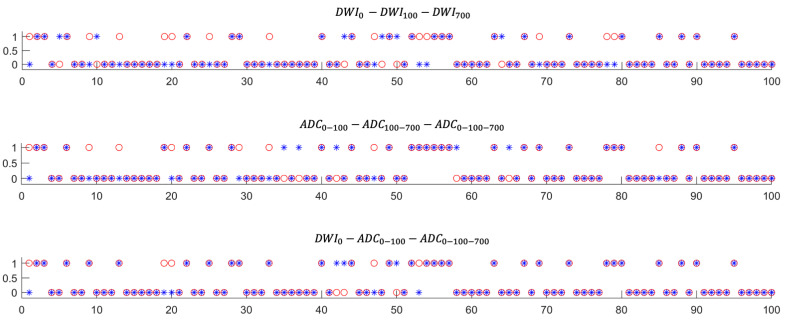
Results of survival prediction model according to the image input dataset. In the graphs, the X-axis indicates 100 patients and the Y-axis shows the result of each patient; 0: survival, 1: death, *: prediction of model, and ○: ground truth. The blank circles (○) indicate failed prediction of survival. On the other hand, the filled circles with asterisks indicate correct prediction. By adding ADC information to DWI as an input datum, the number of correct predictions increased, as shown in each individual case.

**Table 1 diagnostics-13-02555-t001:** Patient demographics (*n* = 100).

Characteristics	N	Characteristics	N
Age, years (mean, range)	62 (40–79)	Tumor size (mean, range)	37 (7–90)
Gender (%)		Surgical stage (%)	
Male	57 (57)	IA	30 (30)
Female	43 (43)	IB	27 (17)
Smoking (%)		IIA	12 (12)
Never-smoker	44 (44)	IIB	8 (8)
Ex-smoker	40 (40)	IIIA	14 (14)
Current smoker	16 (16)	IIIB	2 (2)
Pack-year (mean, range)	21.4 (0–150)	IV	7 (7)
Pathology (%)		Prognosis (%)	
Adenocarcinoma	63 (63)	Death	34 (34)
Squamous cell carcinoma	32 (32)	Survival	66 (66)
Adenosquamous carcinoma	2 (2)	Progression (%)	
Large cell neuroendocrine carcinoma	1 (1)	Progression-free	63 (63)
Pleomorphic carcinoma	1 (1)	Progression	37 (37)
NSCLC, other	1 (1)	Local recurrence	2 (2)
		Metastasis	36 (36)

**Table 2 diagnostics-13-02555-t002:** Survival prediction with deep learning model using DWI with or without ADC.

Input Data	Prediction Results	Survival	Death	AUC ^1^	Kappa ^2^	Sensitivity	Specificity	Accuracy (%)	Balanced Accuracy (%) ^3^	AUC ^1^ Difference (95% CI)
DWI_0_-DWI_0_-DWI_0_	Survival	57	15	0.711	0.441	0.559	0.864	76	71	0.193
	Death	9	19							(0.116, 0.279)
DWI_0_-DWI_100_-DWI_700_	Survival	60	13	0.763	0.554	0.618	0.909	81	76	0.141
	Death	6	21							(0.065, 0.223)
ADC_0-100_-ADC_100-700_- ADC_0-100-700_	Survival	61	8	0.844	0.704	0.765	0.924	87	84	0.061
	Death	5	26							(−0.027, 0.151)
DWI_0_-DWI_700_-ADC_0-100-700_	Survival	63	6	0.889	0.795	0.824	0.955	91	89	0.015
	Death	3	28							(0, 0.048)
DWI_0_-ADC_0-100-700_- ADC_0-100-700_	Survival	63	7	0.874	0.771	0.794	0.955	90	87	0.029
	Death	3	27							(0, 0.074)
DWI_0_-ADC_0-100_-ADC_0-100-700_	Survival	63	5	0.904	0.819	0.853	0.955	92	90	Reference
	Death	3	29							

^1^ AUC, area under curve. ^2^ Kappa, Cohen’s kappa as a measure of the proportion of “true” agreement beyond that expended by chance. ^3^ Balanced accuracy, average of sensitivity and specificity to deal with the class imbalance problems.

**Table 3 diagnostics-13-02555-t003:** Accuracy (%) of survival prediction with regression model incorporating clinical information and AI-generated predictions using diffusion MRI. *: baseline accuracy predicted via deep learning using MRI parameters.

	Baseline Accuracy *	Incorporated Clinical Information						
		Age	Sex	Smoking Pack-Year	Tumor Size	Pathologic Type	Surgical Stage	All
DWI_0_-DWI_0_-DWI_0_	76	78	76	76	77	76	76	83
DWI_0_-DWI_100_-DWI_700_	81	81	81	81	82	81	81	89
ADC_0-100_-ADC_100-700_-ADC_0-100-700_	87	87	87	87	88	87	86	92
DWI_0_-DWI_700_-ADC_0-100-700_	91	91	91	91	92	91	91	94
DWI_0_-ADC_0-100-700_-ADC_0-100-700_	90	90	90	90	91	90	88	93
DWI_0_-ADC_0-100_-ADC_0-100-700_	92	92	92	92	93	92	92	94

## Data Availability

Not applicable.

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
