# Peer review of "Predicting Non-Small-Cell Lung Cancer Survival after Curative Surgery via Deep Learning of Diffusion MRI"

_diagnostics, 2023, doi:10.3390/diagnostics13152555_

Round 1

Reviewer 1 Report

The study evaluated the predictive power of the survival model using deep learning of diffusion-weighted images (DWI) in patients with non-small cell lung cancer (NSCLC). The authors included data from 100 patients and built a good predictive model to predict survival after radical treatment for NSCLC. Here are some suggestions for improving paper.

1.     The prediction model will be better and more convincing if it further verified by other hospital data.

2.     In the part of “3.3. Performance of the survival prediction model using DWI, ADC, and clinical information”: A table is suggested to show the performance of DWI and ADC combined with basic clinical information in predicting survival, more than accuracy, to make the results more intuitive like Table 2.

3.     The authors only used DWI as one indicator, in fact, the degree of lesions enhancement may also affect the prognosis, and therefore, we suggest to include more imaging indicators in subsequent studies.

4.     In line 220-221, I guess you may have made a clerical mistake. Please check whether the figures are correct.

5.     Is it necessary for several doctors to participate in the segmentation of lesions to ensure the authenticity and reliability of data?

6.     The amount of data is relatively small, and the sample size should be increased in subsequent studies.

7.     Were the clinical data included by the authors sufficient? Should postoperative immunohistochemical indicators be included in the study?

The sentence of the article is smooth and the meaning is conveyed accurately.

Author Response

Reviewer #1

Comments and Suggestions for Authors

The study evaluated the predictive power of the survival model using deep learning of diffusion-weighted images (DWI) in patients with non-small cell lung cancer (NSCLC). The authors included data from 100 patients and built a good predictive model to predict survival after radical treatment for NSCLC. Here are some suggestions for improving paper.

1-1. The prediction model will be better and more convincing if it further verified by other hospital data.

  • Thank you for your comments. Collecting data for survival prediction research can be challenging, especially considering the required 5-year time span. So, it was difficult to verify with data from other hospitals in this study. However, we have a plan to conduct a follow-up study and collaborate with other hospitals for external verification.

1-2.     In the part of “3.3. Performance of the survival prediction model using DWI, ADC, and clinical information”: A table is suggested to show the performance of DWI and ADC combined with basic clinical information in predicting survival, more than accuracy, to make the results more intuitive like Table 2.

Reference

Incorporated clinical information

Age

Sex

Smoking pack-year

Tumor size

Pathologic type

Surgical stage

All

DWI0-DWI0-DWI0

DWI0-DWI100-DWI700

ADC0-100-ADC100-700-ADC0-100-700

DWI0-DWI700-ADC0-100-700

DWI0-ADC0-100-700-ADC0-100-700

DWI0-ADC0-100-ADC0-100-700

76

(57/19)

81

(60/21)

87

(61/26)

91

(63/28)

90

(63/27)

92

78

(54/24)

81

(56/25)

87

(61/26)

91

(64/27)

90

(64/26)

92

76

(54/22)

81

(57/24)

87

(60/27)

91

(63/28)

90

(63/27)

92

76

(52/24)

81

(55/26)

87

(62/25)

91

(60/31)

90

(62/28)

92

77

(53/24)

82

(54/28)

88

(63/25)

92

(62/30)

91

(64/27)

93

76

(52/24)

81

(54/28)

87

(63/24)

91

(61/30)

90

(64/26)

92

76

(53/23)

81

(59/22)

86

(63/23)

91

(62/29)

88

(62/26)

92

83

(57/26)

89

(59/30)

92

(64/28)

94

(63/31)

93

(63/30)

94

(63/29)

(62/30)

(63/29)

(61/31)

(63/30)

(59/33)

(63/29)

(63/31)

  • Thank you for your comments. Based on your advice, we can demonstrate Table 3 as following; the numbers in parentheses indicate the count of individuals who survived and the count of individuals who died (Survivors/Deceased).
  • Actually Table 3 is for intuitively showing just increased accuracy by adding clinical information to MRI parameters only models. Thus the ‘Reference’ in the table is not the meaning of gold standard, but the accuracy result of MR parameters. We understand that can cause some confusion and are sorry for the table could be seen not so intuitively. Thanks to your comment, we changed the term ‘Reference’ to ‘Baseline accuracy’ and added annotation on the script and we think it’s enough for increasing immediacy of the table.

Baseline accuracy*

Incorporated clinical information

Age

Sex

Smoking pack-year

Tumor size

Pathologic type

Surgical stage

All

DWI0-DWI0-DWI0

DWI0-DWI100-DWI700

ADC0-100-ADC100-700-ADC0-100-700

DWI0-DWI700-ADC0-100-700

DWI0-ADC0-100-700-ADC0-100-700

DWI0-ADC0-100-ADC0-100-700

76

81

87

91

90

92

78

81

87

91

90

92

76

81

87

91

90

92

76

81

87

91

90

92

77

82

88

92

91

93

76

81

87

91

90

92

76

81

86

91

88

92

83

89

92

94

93

94

Table 3. Accuracy(%) of survival prediction with regression model incorporating clinical information and AI-generated predictions using diffusion MRI. *: baseline accuracy predicted with deep learning using MRI parameters.

1-3.     The authors only used DWI as one indicator, in fact, the degree of lesions enhancement may also affect the prognosis, and therefore, we suggest to include more imaging indicators in subsequent studies.

  • Thank you for the good comment. We understand and agree with you that the extent of enhancement can indicate the prognosis of lung cancer because it reveals tumor angiogenesis and microvessel density, thus treatment response can be predicted via enhancement. As the same reason as #1-1, we cannot repeat the MR examination including dynamic enhancement image protocols for the same patient group, but we will consider this point and gain the images in our succeeding study.

1-4.     In line 220-221, I guess you may have made a clerical mistake. Please check whether the figures are correct.

  • Thank you for your comment. There was a patient who showed both local recurrence and metastasis, thus total number of patients with progression was 37. We add indentations at the paragraph of table, to prevent confusion.

.

1-5.     Is it necessary for several doctors to participate in the segmentation of lesions to ensure the authenticity and reliability of data?

  • Thank you for your comment. In this study, a radiologist specializing in pulmonary imaging with over 20 years of experience was consulted during the lesion segmentation process. Furthermore, the consensus of radiologists with more than 10 years of experience in lung imaging was obtained to confirm the results.

1-6.     The amount of data is relatively small, and the sample size should be increased in subsequent studies.

  • Thank you for your comment. As mentioned earlier, collecting data for survival prediction research over a 5-year time span was indeed challenging. However, we acknowledge that the dataset may be considered relatively small for deep learning research, despite our efforts to address this limitation through augmentation techniques. Therefore, we are fully committed to gathering as much data as possible for future research endeavors.

1-7.     Were the clinical data included by the authors sufficient? Should postoperative immunohistochemical indicators be included in the study?

  • Thank you for your comments. Several immunohistochemical indicators such as EGFR, VEGFR, Ki‐67, p53 and Bcl‐2 are suggested as potential prognostic markers, and we recognize their promising possibility of standard for the differentiation of lung cancer treatment group. But the exact clinical role of these proteins as prognostic marker is still on debates and requires further investigations. We didn’t include the markers expression in this study, but we consider analysis of immunohistochemical indicators as prognosis precursor in our subsequent studies.

Reviewer 2 Report

I have the following comments:

-Lines 51-52. The strengths, weaknesses and potential applications of MRI for the assessment of lung cancer should be illustrated more accurately and in more detail. It would also be advisable to explain the specific role of MRI in an application such as lung cancer staging, for which multidetector CT is the mainstay imaging modality. 

Moreover (down to line 65), it would be important to also stress not only the advantages, but also the potential limitations of DW-MRI for cancer staging (see e.g. Moreira ASL et al, doi 10.3390/jpm13050728).

-Lines 69-72. CNN is not properly a branch of deep learning, but a class of methods that can be successfully applied to image analysis, as explained in the following text.

-Lines 78-81. I suggest removing this part here and merging/integrating it to the final paragraph, wherein the study purpose is described in full.

-Lines 156-157. Please remove "from the University of Oxford".

Author Response

Reviewer #2

Comments and Suggestions for Authors

I have the following comments:

2-1.      -Lines 51-52. The strengths, weaknesses and potential applications of MRI for the assessment of lung cancer should be illustrated more accurately and in more detail. It would also be advisable to explain the specific role of MRI in an application such as lung cancer staging, for which multidetector CT is the mainstay imaging modality.

  • Thank you for the nice comments. We added information of MRI for lung cancer as recommended, as following:
  • Although the ability of pulmonary nodule detection is lower than CT, MRI can detect malignant nodules without radiation exposure. In TNM staging of lung cancer, MRI shows superiority in the evaluation of mediastinal or chest wall invasion, or Pancoast tumor, and comparable N staging assessment capability with PET/CT. Use of diffusion weighted images (DWI) achieves good results of detection and characterization of malignant nodule or lymph node metastasis.

2-2.      Moreover (down to line 65), it would be important to also stress not only the advantages, but also the potential limitations of DW-MRI for cancer staging (see e.g. Moreira ASL et al, doi 10.3390/jpm13050728).

  • Thank you for the balanced comment. We added phrase of limitation as recommended as following;
  • Whereas there is also a report with suspicion in usefulness of ADC values in histological grade of malignancy, despite of the excellence of MRI staging in endometrial cancer.

2-3.      -Lines 69-72. CNN is not properly a branch of deep learning, but a class of methods that can be successfully applied to image analysis, as explained in the following text.

  • This is a good point. We changed the term as recommended.

2-4.      -Lines 78-81. I suggest removing this part here and merging/integrating it to the final paragraph, wherein the study purpose is described in full.

  • Thank you for the edition point. We removed and merged the part as recommended.

2-5.      -Lines 156-157. Please remove "from the University of Oxford".

  • Thank you for the nice comment. We removed terms ‘from the University of Oxford’ as recommended.

Round 2

Reviewer 1 Report

Authors gave good answers to the questions raised. The quality of manuscripts has been improved.